# Amplitude Normalization for Speed-Induced Modulation in Rotating Machinery Measurements

**DOI:** 10.3390/s25206374

**Published:** 2025-10-15

**Authors:** Zhiwen Fang, Qing Zhang, Xinfa Shi

**Affiliations:** 1School of Instrument Science and Technology, Xi’an Jiaotong University, Xi’an 710049, China; fangzw@stu.xjtu.edu.cn; 2Guangzhou Mechanical Engineering Research Institute Co., Ltd., Guangzhou 510700, China; shixinfa@gmeri.com

**Keywords:** fault detection, varying speed condition, amplitude normalization, support vector regression

## Abstract

Rotating machinery under variable-speed conditions suffers from amplitude modulation (AM) effects induced by speed fluctuations, complicating accurate fault detection. To address this issue, an amplitude normalization method based on support vector regression (SVR) is proposed to estimate and remove the AM effects. The method employs a correlation-based feature selection strategy to construct feature vectors strongly associated with rotational speed, thereby enabling the accurate quantification of speed-induced AM effects. The robust nonlinear fitting capability of SVR is then utilized to model and remove these effects, enhancing fault signal clarity. The proposed method is validated through two case studies and compared with advanced amplitude normalization techniques, demonstrating its superior accuracy, robustness, and reliability. Experimental results demonstrate that the proposed method accurately estimates and eliminates speed-induced AM, significantly improving fault diagnosis accuracy by up to 34.7%.

## 1. Introduction

Rotating machinery plays a vital role in industrial systems, where vibration signal-based condition monitoring and fault diagnosis are widely adopted for detecting mechanical faults [1,2,3]. However, variations in operating speed often introduce significant modulation effects into the vibration signals, including amplitude modulation (AM) and frequency modulation (FM). These speed-induced modulations can be confused with fault-induced features, complicating the distinction between operational condition transfer and true degradation of machine health [4]. If not properly addressed, speed-related modulation effects may result in false alarms or missed detections, compromising the reliability and accuracy of condition monitoring systems. Therefore, mitigating the influence of speed variation is crucial for achieving robust and accurate fault diagnosis under variable-speed conditions [5,6,7].

In the past two decades, extensive research has been devoted to condition monitoring under variable-speed operation [8,9,10], with a particular focus on modulation effects in vibration signals. FM is often addressed using order tracking techniques [11,12,13], and numerous studies have demonstrated the effectiveness of computed order tracking in suppressing FM. However, AM effects caused by speed fluctuations remain a challenging and relatively underexplored issue. This study aims to fill this gap by focusing on AM, which may distort amplitude-related fault features and lead to misdiagnosis if not properly compensated.

The amplitude of a vibration signal may vary due to speed fluctuations, faults, or both. When attempting to remove speed-induced AM effects, there is a risk of also attenuating fault-induced amplitude features, potentially masking critical diagnostic information [14]. Hence, the accurate estimation and compensation of speed-induced AM is essential for reliable fault identification. Existing approaches for addressing this issue primarily include envelope-based analysis and speed correlation-based normalization methods.

Envelope analysis-based methods estimate AM effects by extracting the envelope of the vibration signal or its components to capture amplitude variations induced by speed fluctuations. Urbanek et al. [15] divided the original signal by its estimated envelope to remove the AM effect. Stander et al. [16] applied a low-pass filter to the absolute value of the analytic signal and normalized the result using the filtered envelope. Schmidt et al. [4] employed a moving median filter on the squared envelope and used the root mean square (RMS) of the median envelope for normalization. Despite differences in implementation, these methods all rely on a single amplitude metric to estimate the speed-induced AM effect, which is insufficient to fully capture its complexity. Moreover, fault-related information is often embedded in the envelope and may be inadvertently suppressed during normalization. To mitigate this issue, several recent studies have proposed refined approaches. For instance, Zhang et al. [17] extracted the signal envelope, applied a low-pass filter with a cutoff frequency based on the maximum speed variation rate to preserve only the speed-related low-frequency components, and then normalized the signal using the filtered envelope. This method aims to retain fault information while eliminating speed-induced AM.

Alternatively, rotational speed-correlated function-based methods estimate AM effects by constructing mathematical models that link amplitude variation to speed changes. Wei et al. [18] modeled the AM effect as proportional to the square of the speed (s^2^) and normalized the signal accordingly. However, due to resonance and energy transmission losses, the actual relationship between speed and AM is often more complex than a simple squared function. Rao et al. [14] addressed this by fitting a piecewise power function, but the approach suffers from limited expressiveness, sensitivity to noise, and poor performance at boundary regions, making it inadequate for capturing highly nonlinear AM behaviors. More recently, with the rapid development of deep learning, researchers have also attempted to tackle this problem using neural networks. For example, Rao et al. [19] employs a convolutional neural network to learn a nonlinear speed normalization function in a data-driven manner. While such approaches show potential in capturing complex AM behaviors, they generally require large training datasets, involve heavy computational cost, and suffer from limited interpretability, which may restrict their applicability in practical condition monitoring systems.

To overcome these limitations, this paper proposes an amplitude normalization model based on support vector regression (SVR) to accurately estimate and eliminate speed-induced AM effects. The model leverages the nonlinear regression capability of SVR to learn the complex relationship between rotational speed and AM. A speed correlation analysis strategy is introduced to extract relevant features, with feature importance used for weighted modeling to ensure balanced influence on the fitting process. Once trained, the model is applied to normalize the vibration signals, effectively removing speed-induced AM while preserving fault-related information. The normalized signals are then used for fault detection.

The proposed method is validated using two distinct datasets, demonstrating its superior performance in both AM suppression and diagnostic accuracy. Unlike traditional per-signal processing approaches such as manual envelope extraction, the SVR-based model enables offline training and real-time deployment without repeated tuning, making it well suited for scalable, condition-based monitoring of rotating machinery under variable-speed conditions.

The main contributions of this paper are summarized as follows:
First application of machine learning to model speed–AM relationships: This study introduces an SVR-based amplitude normalization method that, for the first time, employs machine learning to capture the nonlinear mapping between rotational speed and AM. Unlike traditional envelope extraction or simple functional approximations, the proposed model leverages SVR’s nonlinear fitting capability and robustness, while being trained exclusively on healthy-state data to avoid suppressing fault-related features.Multi-feature fusion strategy for AM quantification: To address the limitations of single-feature estimation, we propose a correlation-guided framework that constructs a multi-feature set with strong relevance to rotational speed. This fusion strategy allows for a more accurate and comprehensive characterization of AM effects, improving the precision of normalization.Enhanced fault detection under variable-speed conditions: By integrating the SVR-based normalization model with the multi-feature quantification strategy, the method effectively suppresses speed-induced AM effects. This leads to significantly improved fault detection accuracy and reliability, demonstrating the practical value of the approach in industrial applications.

The remainder of this paper is organized as follows: Section 2 presents the proposed AM effect measurement method based on multi-feature fusion. Section 3 details the SVR-based amplitude normalization model. Section 4 provides experimental validation using two representative datasets. Finally, Section 5 concludes the paper with a summary of findings and a discussion of the study’s limitations and future work.

## 2. Theoretical Background

In this section, the principles of AM effects caused by speed variation are explained in detail. Existing methods for quantifying AM effects are discussed, and their limitations are emphasized.

### 2.1. The Amplitude Modulation Induced by Speed Variation

In Schmidt’s study [4], a vibration signal x(t) under varying conditions is modeled as the product of a stationary fault-related component r(t) and a modulation function M(t) that reflects operating conditions. When the rotational speed ω(t) fluctuates, parameters such as excitation energy and impact intensity vary accordingly, making M(t)=f(ω(t)). This relationship can be expressed as(1)x(t)=r(t)⋅M(t)=r(t)⋅f(ω(t))

The modulation function M(t) introduces a low-frequency AM on r(t), as illustrated in Figure 1. In this work, AM effects are considered to arise solely from speed variations, while other factors (e.g., load, sensor transmission) are assumed to be constant.

Essentially, the modulation function approximates the trend of the vibration signal’s amplitude variation. Envelope analysis-based AM estimation methods extract the signal envelope or that of its components to characterize this trend, which then serves as the entity of modulation function. By dividing the original signal by it, the AM effect is attenuated, making the modulation function equivalent to a normalization function. However, applying envelope extraction and normalization to each individual signal in practical monitoring systems is computationally intensive and thus unsuitable for real-time or large-scale industrial applications.

To improve efficiency, speed-correlated methods approximate the AM effect as a predefined function of speed, i.e., $M_{i}=f(\omega_{i})$. $M_{i}$ is the corresponding modulation function, which is a specific envelope form derived through envelope analysis of the vibration signal, and $\omega_{i}$ denotes the associated speed signal. Once this mapping is trained, the modulation can be directly estimated from the speed signal and applied for normalization. To prevent the normalization process from suppressing fault-related information, the normalization function should be trained exclusively on data collected under healthy operating conditions.

### 2.2. Speed-Correlated Function Based Methods and Their Limitations

As outlined in Section 2.1, speed-correlated, function-based methods are among the most effective approaches for estimating AM effects, and are particularly suitable for engineering applications. These methods aim to construct a normalization function by modeling the relationship between rotational speed and the AM effect under healthy operating conditions. Once such a function is established, it can be reused throughout the service life of the equipment without retraining, thus enabling real-time AM suppression and improving fault detection performance.

Existing studies typically implement this framework in two ways. The first relies on single-feature estimation, where one time-domain metric such as RMS is calculated within a moving window and then correlated with speed. While straightforward, this approach fails to fully characterize the dynamic nature of amplitude modulation, as a single feature cannot capture the complete temporal structure of amplitude variation. The second line of work approximates the speed–AM relationship using simple mathematical models, such as squared speed terms or power functions. However, these formulations lack the flexibility to represent local variations and nonlinearities, which are common in practice.

More recently, deep learning models such as autoencoders have been introduced to learn this mapping in a data-driven manner. Although they show potential, such methods usually require large training datasets, involve heavy computational cost, and lack interpretability, which limits their industrial applicability.

In summary, accurate estimation of AM effects under variable speed requires overcoming the limitations of single-feature metrics, simplistic functional approximations, and computationally intensive deep models. This motivates the SVR-based normalization model proposed in this paper, which combines nonlinear regression capability with efficiency and robustness for practical use.

## 3. Proposed Amplitude Normalization Model

### 3.1. Measurement of AM Effect Based on Multi Feature Fusion

Vibration signals contain numerous features, not all of which are relevant for accurately measuring AM effects. To improve measurement accuracy, this study proposes a feature selection strategy based on speed correlation, enabling the identification of features that are highly indicative of AM effects while excluding irrelevant ones. The overall multi-feature fusion strategy for AM effect measurement is illustrated in Figure 2.

Commonly used features in vibration signal analysis include the median, RMS, peak-to-peak value, and kurtosis. In this study, these four features are selected to validate the effectiveness of the proposed multi-feature fusion method. While other features may also contribute, they are not considered here, as the primary objective is to demonstrate the feasibility and core advantages of the proposed approach. The inclusion of additional features will be explored in future work to further improve the comprehensiveness and accuracy of AM effect estimation.

A limitation of existing methods is the absence of effective criteria for evaluating whether a feature can reliably represent AM effects. To address this gap, this study adopts the Pearson correlation coefficient between each feature and rotational speed, referred to as the Feature–Speed Pearson Correlation (FSPC), as a criterion for feature selection. This coefficient quantitatively evaluates the degree to which a feature correlates with speed variation, thereby serving as a proxy for its sensitivity to speed-induced AM effects. A higher FSPC value indicates a stronger correlation, implying that the feature is more likely to capture the modulation effect driven by speed fluctuations:(2)$$FSPC_{y_{i}}=corr(s,y_{i})=\frac{\sum \left(s-\bar{s}\right)(y-{\bar{y}}_{i})}{\sqrt{\sum {\left(s-\bar{s}\right)}^{2}\sum {\left(y-{\bar{y}}_{i}\right)}^{2}}}$$
where s represents the speed sequence which can be obtained by discretizing the continuous speed signal ω(t), and $y_{i}$ represents a certain feature sequence. $\bar{x}$ and $\bar{y_{i}}$ are the average values of speed sequence and the certain feature sequence.

To construct an effective supervised learning model, features with low correlation to speed-induced AM effects must be excluded. Specifically, features with a correlation coefficient below a predefined threshold are removed from the dataset. The remaining features are then assigned weights proportional to their FSPC values. This weighting strategy enhances the contribution of highly correlated features, thereby improving the accuracy and robustness of AM effect measurement. By integrating the weighted features, the proposed approach achieves a more comprehensive representation of speed-induced AM effects, enhancing the reliability of subsequent signal normalization.

### 3.2. Struct of Amplitude Normalization Model

SVR is an extension of Support Vector Machines tailored for regression analysis which is employed to model the nonlinear relationship between speed and the corresponding AM effects. Owing to its strong nonlinear fitting capability and adaptability to complex functional relationships [20,21], SVR is well-suited for this task. While alternative models could also be used to address such problems, this study focuses on validating the effectiveness of the SVR-based amplitude normalization framework. Once validated, future research can investigate alternative implementations to further improve model performance and adaptability. A flowchart of the proposed normalization model is presented in Figure 3.

The amplitude normalization model achieves AM effect estimation and removal through four main steps:

(1) Data collection: Under healthy operating conditions, synchronized vibration and speed signals are collected from the rotating machinery. All healthy-state data are used for training to ensure that the recorded speed range comprehensively covers the full operational range of the equipment.

(2) Measurement of AM effect: This critical step involves feature extraction, data preprocessing, correlation analysis, and feature weighting. Specifically, four time-domain features, i.e., median, RMS, peak-to-peak value and kurtosis, are extracted from the vibration signal using a sliding window of predefined size and overlap. The resulting feature vectors are denoted as $y_{1}$, $y_{2}$, $y_{3}$, and $y_{4}$, respectively. A moving average window of the same configuration is applied to the speed signal to obtain the speed feature vector s. The windowing parameters follow the recommendations in Ref. [15] to ensure consistency with prior studies. In this study, the window length is set to 1/500 of the total signal duration. This choice ensures that the window is long enough to capture fluctuations in the vibration signal, while remaining short enough for the speed within each window to be considered approximately stable.

To ensure data quality, preprocessing is conducted following feature extraction. This includes outlier removal, imputation of missing values, and feature normalization. Additionally, a phase compensation algorithm is applied to both the vibration and speed signals to mitigate the effects of phase shifts. All feature vectors are then scaled to the range [1] using min–max normalization. 

Next, the correlation between each feature and the speed vector is quantified using the FSPC metric, as defined in Equation (2). Features with higher FSPC values are more strongly associated with speed variation and therefore better reflect the underlying AM effects. In contrast, only features with positive FSPC values greater than 0.5 are retained. A threshold of 0.5 is widely regarded in the statistical literature as indicating a moderate-to-strong correlation [22]. Negative correlations are excluded, since they do not conform to the physical mechanism of speed-induced amplitude modulation, where vibration amplitude generally increases with rotational speed.

Finally, for each retained feature vector, a corresponding weight is assigned based on its FSPC value. Features more strongly correlated with speed exert greater influence in the final AM effect representation. The resulting composite feature vector y, which quantitatively characterizes the AM effect, is computed as follows:(3)$$y=\sum_{i=1}^{N}\omega_{i}y_{i}$$
where $y_{i}$ represents the i-th feature vector, $\omega_{i}$ is the weight assigned to $y_{i}$ based on its FSPC, and N is the total number of retained features. By overcoming the limitations of single-feature approaches, the proposed multi-feature fusion strategy ensures that the final feature vector y serves as a comprehensive representation of the AM effect. This vector is subsequently used as the target output in model training, enabling the learning of a nonlinear mapping between speed and AM effects. The learned mapping constitutes the normalization function used to eliminate speed-induced AM effects.

(3) Model training: The basic SVR model is chosen as the learning model for this study. The feature vector $y=\left[y_{1},y_{2},\ldots ,y_{n}\right]\in R^{n}$ obtained in Step 2, is used as the target output, while the corresponding speed vector $s=\left[s_{1},s_{2},\ldots ,s_{n}\right]\in R^{n}$ is used as input. The goal is to learn a normalization function $f\left(s,\theta \right)$ parameterized by θ={ω,b}, where w and b denote the SVR weights and biases, respectively. The regression function is defined as(4)f(s,θ)=wΦ(s)+b
where $\Phi \left(\cdot \right)$ is a nonlinear mapping function to a high-dimensional feature space. The optimal parameters are obtained by minimizing the following objective function,(5)$$\underset{\theta }{\min}\{\frac{1}{2}{\left\|w\right\|}^{2}+C\sum_{i=1}^{N}{\left|f(s_{i})-y_{i}\right|}_{\varepsilon }\}$$
where N is the number of training samples, C is the regularization parameter, $\left\|\cdot \right\|$ is the L2 norm, and ε represents the maximum permissible regression error. The ε-intensive loss function is defined as(6)$${\left|f(s_{i})-y_{i}\right|}_{\varepsilon }=\max\{0,\left|f(s_{i})-y_{i}\right|-\varepsilon \}$$

With the addition of the relaxed variables $\xi_{i}$ and $\xi_{i}^{*}$, the optimization problem for the target function is formulated as(7)$$\begin{array}{l} \min\{\frac{1}{2}{\left\|w\right\|}^{2}+C\sum_{i=1}^{N}\left(\xi_{i}+\xi_{i}^{*}\right)\}\\ s.t.=\left\{\begin{array}{l} y_{i}-f(s_{i})\leq \varepsilon +\xi_{i}\\ f(s_{i})-y_{i}\leq \varepsilon +\xi_{i}^{*}\\ \xi_{i},\xi_{i}^{*}\geq 0 \end{array}\right. \end{array}$$
where $\xi_{i}$ and $\xi_{i}^{*}$ represent the slack variables for the i-th training sample. Solving the dual problem using Lagrange multipliers and applying the kernel trick yields the final regression function,(8)$$f(s)=\sum_{i=1}^{N}\left(\alpha_{i}-\alpha_{i}^{*}\right)K(s_{i},s_{j})+b$$
where $\alpha_{i,}\alpha_{i}^{*}$ are Lagrange multipliers, K(·,·) is the kernel function, while $s_{i}$ and $s_{j}$ are components of the input vector s. Given the nonlinear nature of the relationship, a radial basis function (RBF) kernel is employed,(9)$$K(s_{i},s_{j})=\exp(-\gamma {\left\|s_{i}-s_{j}\right\|}^{2})$$
where γ denotes the width of the RBF kernel function. The performance of the SVR model, including its estimation accuracy and fitting ability, is significantly determined by key parameters such as the regularization parameter C, the kernel coefficient γ, and the margin of tolerance ε. For simplicity, $\Lambda =\left\{C,\gamma ,\varepsilon \right\}$ is used to represent the trainable parameter set of SVR model. The model is trained by minimizing the mean squared error between the output $f\left(s\right)$ and the target output y, which is described as(10)$$L(f(s),y)=\frac{1}{n}\sum_{i=1}^{N}{\left(f(s_{i})-y_{i}\right)}^{2}$$(11)$$\Lambda^{*}=\arg\underset{\Lambda }{\min}L(f(s;\Lambda ),y)$$
where $\Lambda^{*}$ is the optimized set of hyperparameters. The process of parameter optimization is so-called model training.

The commonly used parameter optimization methods include grid search [23], Bayesian optimization [24], particle swarm optimization [25,26], and genetic algorithm [27,28]. In this study, the widely used grid search method is employed to select the parameters of the SVR model for simplicity. Although there are more advanced hyperparameter optimization techniques, the focus of this study is not on hyperparameter optimization. Once the proposed amplitude normalization model is proved to be effective, other parameter optimization methods should also be applicable. The details of parameter optimization using grid search can be found in Ref. [21]. The parameter grid for amplitude normalization model is defined as $C=\left\{0.1,1,10,100\right\}$, $\gamma =\left\{0.1,1,10\right\}$, and $\varepsilon =\left\{0.01,0.1,1\right\}$.

(4) Normalization of vibration signal: The well-trained amplitude normalization model, with optimal parameters, is applied to normalize either healthy or faulty signal to eliminate the AM effect,(12)$$x_{norm}=\frac{x}{f^{*}(s)}$$
where $x_{norm}$ is the normalized vibration signal with speed-induced AM effects removed, and $f^{*}(s)$ denotes the output of the SVR model using the optimal parameters $\Lambda^{*}$. The normalized signal is subsequently used for intelligent fault diagnosis under variable-speed conditions. As the AM interference has been effectively suppressed, the resulting signal is expected to improve diagnostic accuracy.

In summary, this study reframes the estimation of speed-induced AM effects as a nonlinear regression problem based on a speed-correlated function framework. Conventional methods often suffer from two key limitations: incomplete characterization of AM effects due to reliance on single-feature metrics, and insufficient modeling of the nonlinear relationship between speed and modulation. These shortcomings lead to inaccurate or suboptimal normalization performance.

To address these challenges, the proposed amplitude normalization model integrates the multi-feature fusion strategy for more comprehensive AM effect quantification and employs the SVR model to learn the complex mapping between speed and AM. By jointly enhancing both the measurement and modeling components, the method provides a more accurate and robust solution for eliminating speed-induced AM effects while preserving fault-related information.

## 4. Case Studies

In this section, the effectiveness of the proposed amplitude normalization model in estimating and eliminating AM effects induced by speed variations is validated using two experimental datasets. Additionally, the impact of signal normalization on fault detection performance is evaluated.

### 4.1. Case 1

The dataset for Case Study 1 was collected by authors at Taiyuan University of Technology, Shanxi, China [29]. It comprises bearing operation data under three constant-speed conditions, four variable-speed conditions, and two distinct load levels. The bearing health states are categorized into six types: healthy, inner race fault, outer race fault, rolling element fault, two-fault combinations, and three-fault combinations. This study focuses on the variable-speed dataset, with speeds ranging from 15 Hz to 35 Hz. The experimental setup is illustrated in Figure 4. An accelerometer is mounted directly above the test bearing to capture vibration signals during operation. Meanwhile, the motor speed is monitored in real time using a DC drive and transmitted to the host computer. Vibration signals are sampled at 10 kHz, with each recording lasting 10 s.

#### 4.1.1. Measurement of AM Effect in Case 1

To prevent the amplitude normalization model from inadvertently removing fault-related features, the model is trained exclusively on healthy-state data. Specifically, all available healthy-state signals across the full operating speed range are used as the training set for model development. This ensures that the learned normalization function captures only the speed–AM relationship and remains valid over the entire speed interval. Feature extraction is then performed on this set by segmenting the vibration signals using a non-overlapping sliding window of 0.02 s. From each segment, four time-domain features, median, RMS, peak-to-peak value, and kurtosis, are extracted to construct corresponding feature vectors. Simultaneously, the mean speed of each segment is calculated to produce a speed vector of matching length. The results of feature extraction are presented in Figure 5, and the corresponding FSPC values for each feature are listed in Table 1.

As shown in Figure 5, the median, RMS, and peak-to-peak feature vectors exhibit a consistent upward trend with increasing speed. This aligns with physical expectations, as AM typically intensifies at higher rotational speeds. In contrast, the kurtosis feature remains largely unaffected by speed fluctuations, indicating its limited relevance for characterizing AM effects. The FSPC values support this observation. The first three features demonstrate strong correlations with speed, confirming their suitability for AM effect measurement. However, the kurtosis feature yields an FSPC value of −0.0013, suggesting no meaningful correlation. Accordingly, it is excluded from the feature dataset.

As a result of the speed correlation analysis, the median, RMS, and peak-to-peak feature vectors are selected to quantify the AM effect. This can be expressed as,(13)$$\left\{\begin{array}{l} y=\sum \omega_{i}y_{i}\\ \omega_{i}=\frac{FSPC_{i}}{\sum FSPC_{i}} \end{array}\right.,i=\{Median,RMS,Peak-to-peak\}$$
where $\omega_{i}$ is the weight for each feature. y represents the multi-feature vector used to measure the AM effect, which is further input into SVR model as target output.

#### 4.1.2. Amplitude Normalization Model

The multi-feature vector y serves as the input for training the amplitude normalization model, following the procedure detailed in Section 3.2. The optimal hyperparameter set, obtained Via grid search, is $\Lambda^{*}=\left\{1,0.01,10\right\}$. The corresponding training results are shown in Figure 6. It can be observed that the trained model effectively captures the nonlinear relationship between speed and the AM effect. Notably, the model also successfully identifies resonance-induced amplitude fluctuations at approximately 16 Hz and 32 Hz, as indicated by the dotted lines. These results confirm the capability of the proposed method to accurately estimate speed-induced AM effects.

#### 4.1.3. Normalization of Vibration Signal

To further validate the effectiveness and superiority of the proposed method, a comparative study is conducted with four existing AM effect estimation techniques. These methods are evaluated based on their ability to suppress AM caused by speed variations. A brief overview of each comparative method is provided below:
Schmidt et al. [4] estimate the AM effect by computing the square root of the moving median of the squared signal envelope. The vibration signal is then normalized by dividing it by the estimated AM effect. In this study, we use a 1-s window with 90% overlap for the median operation. After AM estimation, linear interpolation is applied to ensure temporal alignment with the original signal length.Zhang et al. [17] first extract the instantaneous amplitude envelope of the vibration signal, followed by low-pass filtering to retain only the slowly varying component associated with speed fluctuation. The cutoff frequency is determined by the maximum relative speed variation. The signal is then normalized using this low-frequency component.Wei et al. [18] model the AM effect as a quadratic function of rotational speed and remove it by dividing the signal by the squared speed.

These three methods and the proposed method are applied to remove AM effects in signals with inner race faults. Figure 7 displays representative vibration signals before and after normalization. As shown, both methods [4] and [17] yield more stationary signals, primarily due to their envelope-based strategies which smooth amplitude variations. However, as discussed in the Introduction, such approaches risk attenuating not only the speed-induced AM effect but also fault-induced amplitude components, potentially leading to a loss of diagnostic information. This limitation will be further demonstrated in the performance analysis section. The method in [18], which assumes a strict quadratic relationship between amplitude and speed, tends to overestimate the AM effect, leading to pronounced distortion in the normalized signal. In contrast, the proposed method, through nonlinear regression using SVR, accurately models the complex relationship between speed and AM effects. This allows for more precise estimation and selective suppression of speed-induced AM components, while preserving critical fault-related features.

To more clearly illustrate the effectiveness of each normalization method, the health indices, defined as the RMS of the vibration signal with respect to speed, were calculated for both pre- and post-normalization signals. The results are shown in Figure 8. As observed, the method proposed in [4] produces relatively stable health indices across different rotational speeds, indicating effective suppression of speed-induced AM effects. However, the health indicators for healthy and faulty datasets exhibit considerable overlap, suggesting that fault-induced AM components were also attenuated during normalization. This overgeneralization leads to the loss of critical diagnostic information. Similarly, the method in [17] effectively mitigates AM effects. Owing to its low-pass filtering step, it retains primarily the low-frequency components associated with speed variation. Nevertheless, fault-related features are not entirely removed during filtering, resulting in residual fault information within the envelope signal. Consequently, partial overlap between health indices under healthy and faulty conditions remains. In contrast, the method in [18] models the AM effect as a quadratic function of speed, which leads to inaccurate estimations and incorrect health index trends. Despite this modeling error, the distinction between healthy and faulty conditions is preserved due to the retention of fault-related features.

Unlike them, the proposed method achieves both objectives: it yields stable health indices across the entire speed range while maintaining a clear separation between healthy and faulty datasets. This confirms that the model not only captures the nonlinear relationship between speed and AM effects accurately but also preserves essential fault characteristics. These findings further substantiate the effectiveness and superiority of the proposed approach in AM suppression without compromising diagnostic integrity.

#### 4.1.4. Fault Detection

To quantitatively evaluate the contribution of the proposed method to fault detection, a classification framework based on an autoencoder (AE) was employed. The normalized signals obtained from the proposed method and three comparative methods were individually input into the AE model for fault detection. In this stage, the training set consists of both normalized healthy signals and normalized faulty signals, where the faulty class includes inner race, outer race, and mixed fault conditions. The task is therefore defined as binary fault detection (healthy vs. faulty) rather than multi-class fault classification, in line with the practical requirement of determining whether a fault has occurred. To ensure comparability, the data preprocessing and AM model configuration follow the procedure outlined in Ref. [15].

Model training was conducted using the Adam optimizer with an initial learning rate of 0.001, which is the widely recommended default setting for stable convergence [30]. A batch size of 64 was selected as a common compromise between computational efficiency and training stability, and the maximum number of epochs was set to 100 to ensure convergence without overfitting, in line with configurations frequently adopted in autoencoder-based signal processing studies. A threshold of 0.5 was used for classification: outputs above the threshold were labeled as faulty, and those below as healthy. Model performance was evaluated using the area under the receiver operating characteristic curve (AUC), which integrates both true positive and false positive rates to provide a robust performance metric. AUC values range from 0 to 1, with higher values indicating better classification accuracy. The results, presented in Table 2, demonstrate that the proposed method consistently outperforms existing approaches and the baseline scenario without AM effect elimination, thereby confirming its ability to enhance fault detection performance under variable-speed conditions. To ensure reliability, each fault detection experiment was repeated three times, and the reported AUC values represent the average across trials. The corresponding standard deviation values are also reported in Table 2.

### 4.2. Case 2

The dataset for this case study is publicly available from the University of Ottawa [31]. It contains bearing vibration signals collected under various operational conditions, including different rotational speeds, loads, and fault parts such as inner race, outer race, and rolling element. The dataset also incorporates diverse working environments to reflect the complexity of real-world bearing failures. The experimental setup is illustrated in Figure 9.

#### 4.2.1. Measurement of AM Effect in Case 2

Following the procedure described in Section 4.1, only healthy-state data was used for training. Four statistical features, median, RMS, peak-to-peak value, and kurtosis, were extracted from the vibration signal using sliding windows with identical parameters. The results of feature extraction are presented in Figure 10, and the corresponding FSPC values are summarized in Table 3.

As shown in Figure 10, the median and RMS exhibit trends that closely align with speed-induced AM effects. Although the peak-to-peak value increases with speed, it displays significant fluctuations due to its sensitivity to noise. In contrast, kurtosis remains largely unaffected by speed variation and is therefore unsuitable for representing AM effects. The FSPC values further support this assessment: the median and RMS achieve high correlation coefficients (≈0.9), confirming their suitability for AM effect quantification. The peak-to-peak value has a moderate FSPC (0.57), indicating it should be retained but assigned a lower weight to reduce its impact. Kurtosis, with an FSPC of −0.2, is eliminated from the feature set. The final AM effect feature vector is constructed using Equation (13).

#### 4.2.2. Results of Amplitude Normalization Model

The SVR model was trained following the same procedure as in Case 1, and the corresponding results are presented in Figure 11. The optimal parameter set, identified Via grid search, is Λ* = {10, 0.01, 1}. As shown in Figure 11, the trained normalization model accurately captures the trend of the AM effect across varying speeds. This model is subsequently applied to the entire dataset to eliminate speed-induced AM effects.

Figure 12 presents representative normalized vibration signals obtained using the proposed approach, along with those from three existing methods. To quantitatively evaluate normalization performance, Figure 13 plots the RMS of the normalized signals as a function of speed. Consistent with the findings in Case 1, the proposed method demonstrates superior effectiveness in removing AM effects while retaining fault-relevant features.

The normalized data are further preprocessed using the same steps described in Case 1 and subsequently input into the same AE-based fault detection model. As summarized in Table 4, the proposed method yields the best detection performance among all compared methods, further confirming its robustness and effectiveness under complex variable-speed conditions.

## 5. Conclusions and Limitations

This study proposed an amplitude normalization model based on SVR to mitigate AM effects induced by speed variations in vibration signals, aiming to enhance fault detection in rotating machinery. By leveraging the nonlinear fitting capability of SVR and a multi-feature fusion strategy, the proposed method accurately estimates and suppresses speed-induced AM effects while preserving fault-related information. The main findings and contributions are summarized as follows:

First, the proposed method effectively models the nonlinear relationship between rotational speed and AM effects using a feature set strongly correlated with speed. Compared with traditional envelope-based and speed-correlated function-based techniques, the SVR-based model demonstrates improved fitting accuracy and robustness.

Second, the amplitude normalization significantly enhances fault detection performance. Across two case studies, the average AUC improved by 9.0%–34.7% compared to unprocessed data. Relative to existing methods, the proposed approach achieved an AUC increase of 2.9%–27.9%, demonstrating superior ability to suppress AM effects and retain critical fault features under variable-speed conditions.

Despite its promising performance, the proposed method has some limitations. The multi-feature fusion strategy relies on features strongly correlated with speed, which may require refinement for broader applicability across diverse datasets. Moreover, the current framework assumes stable load and sensor conditions, whereas real-world environments may introduce additional sources of modulation. Finally, although the method was validated using both public and lab-collected datasets, these do not fully capture the complexity of industrial operating conditions. Future work will focus on applying the proposed approach to field data from actual machinery to further assess and improve its generalizability and practical utility.

## Figures and Tables

**Figure 1 sensors-25-06374-f001:**
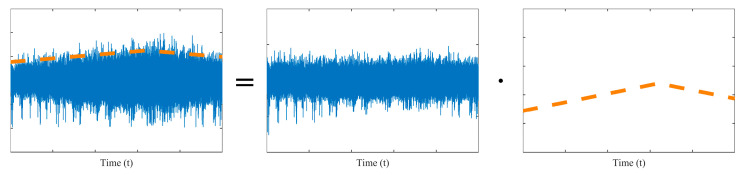
Illustration of the amplitude modulation.

**Figure 2 sensors-25-06374-f002:**
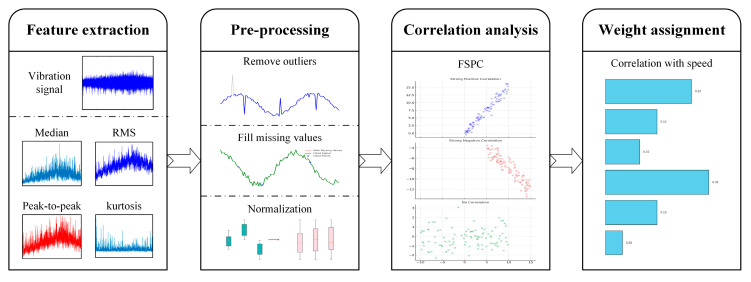
The AM effect measurement strategy.

**Figure 3 sensors-25-06374-f003:**
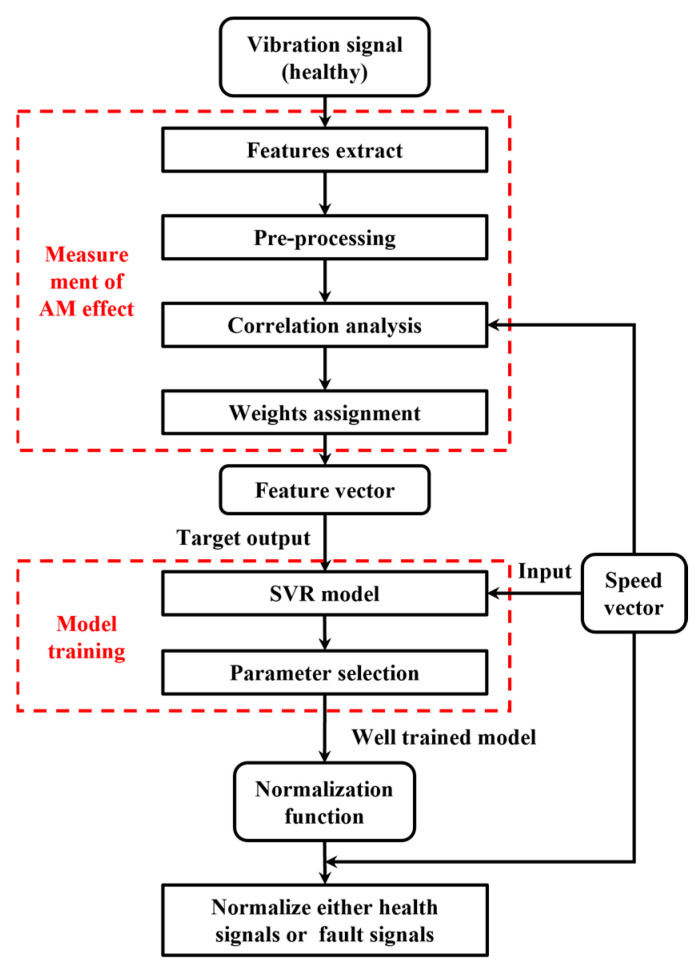
The flowchart of amplitude normalization model.

**Figure 4 sensors-25-06374-f004:**
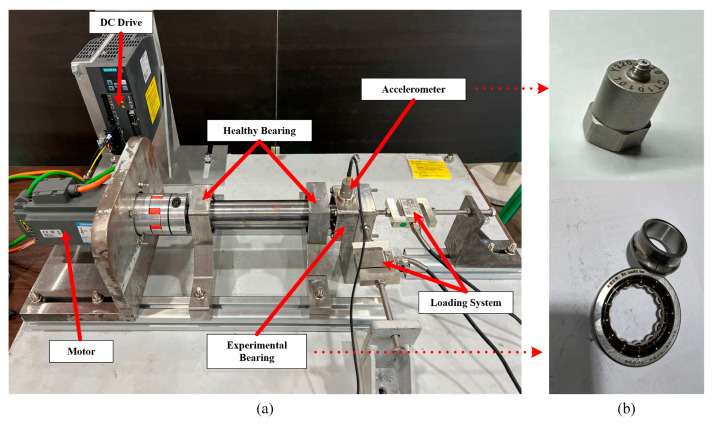
The experimental platform in Ref. [28]. (**a**) device, (**b**) Accelerometer and bearing.

**Figure 5 sensors-25-06374-f005:**
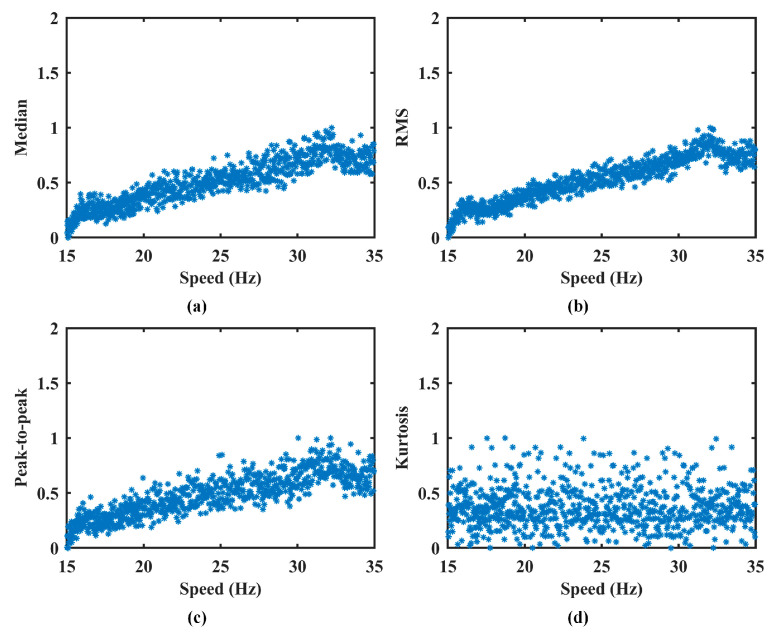
The extracted features in Case 1. (**a**) Median. (**b**) RMS. (**c**) Peak-to-peak. (**d**) Kurtosis.

**Figure 6 sensors-25-06374-f006:**
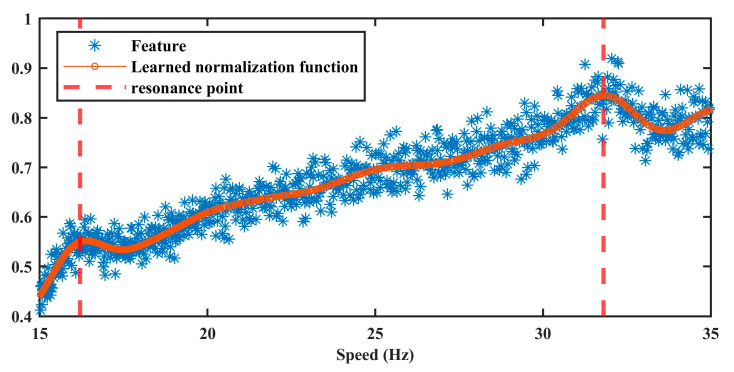
The result of amplitude normalization model in Case 1.

**Figure 7 sensors-25-06374-f007:**
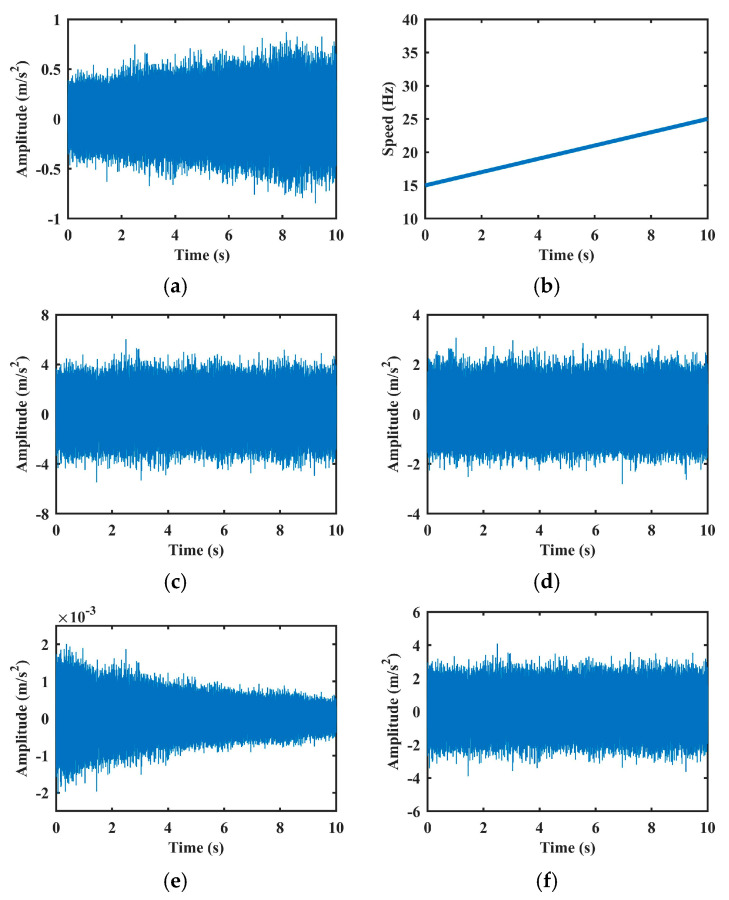
AM effect removal experiments of the vibration signal with inner fault in Case 1: (**a**) Raw vibration signal. (**b**) Rotation speed signal. (**c**–**f**) are results of methods [4,17,18], and the proposed method, respectively.

**Figure 8 sensors-25-06374-f008:**
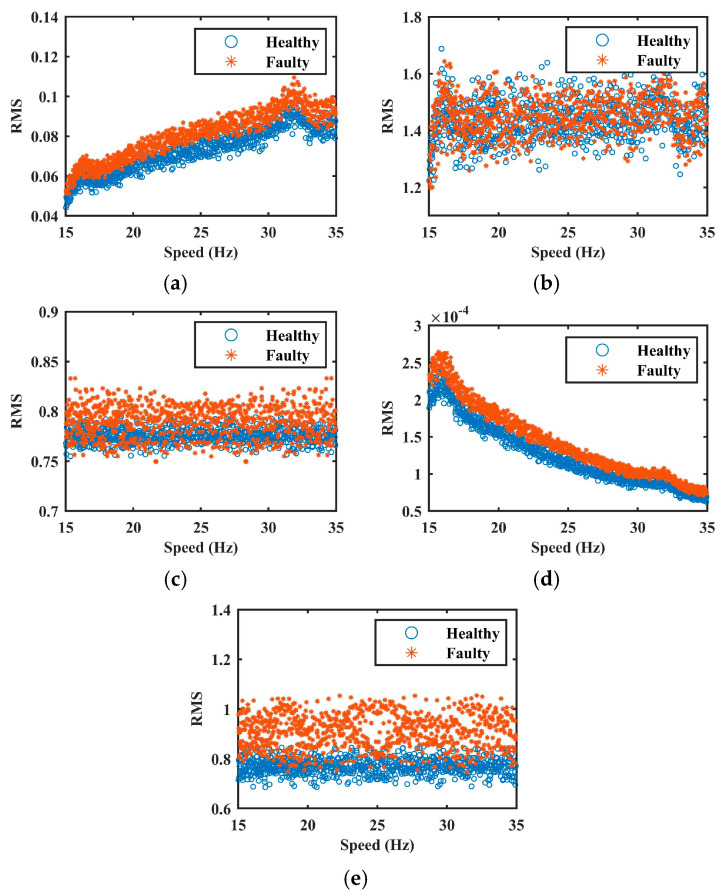
The health indices in Case 1. (**a**) raw signal. (**b**–**e**) are normalized signals by methods of [4,17,18], and the proposed method, respectively.

**Figure 9 sensors-25-06374-f009:**
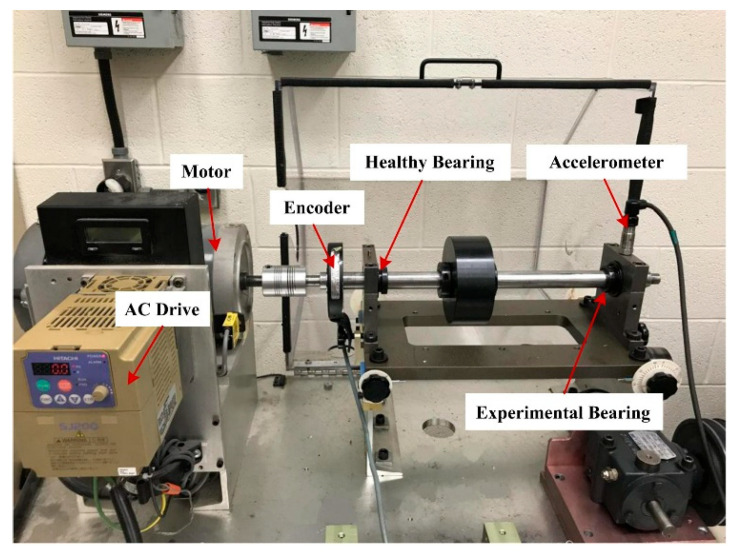
Experiment setup in Ref. [29].

**Figure 10 sensors-25-06374-f010:**
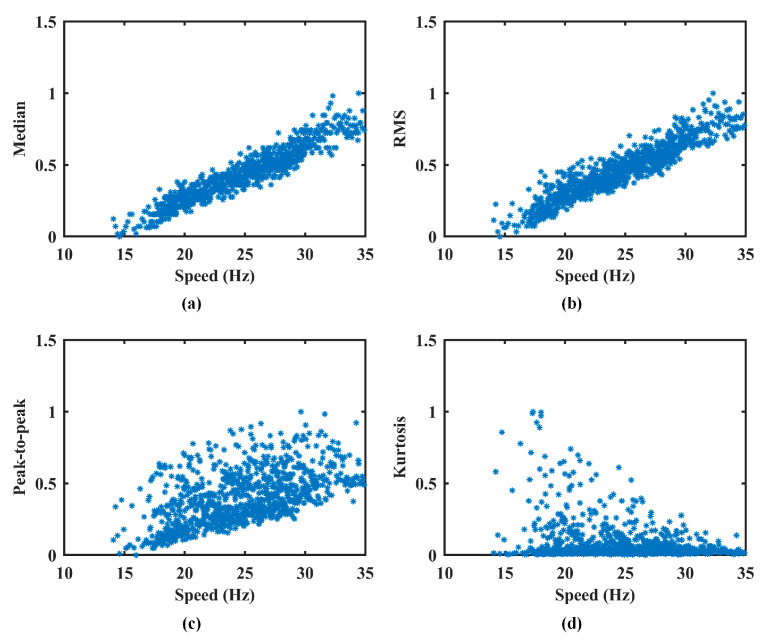
The extracted features in Case 2. (**a**) Median. (**b**) RMS. (**c**) Peak-to-peak. (**d**) Kurtosis.

**Figure 11 sensors-25-06374-f011:**
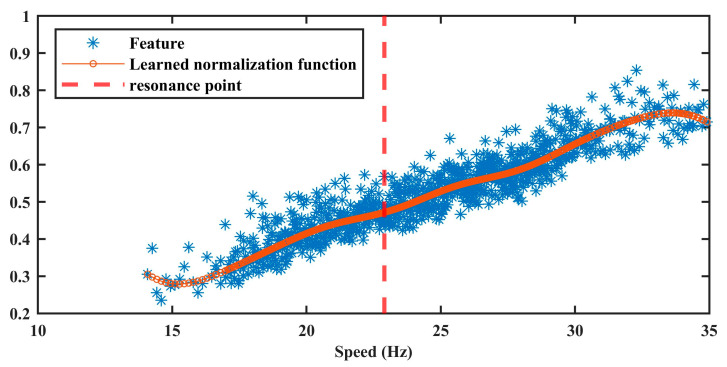
The result of amplitude normalization model in Case 2.

**Figure 12 sensors-25-06374-f012:**
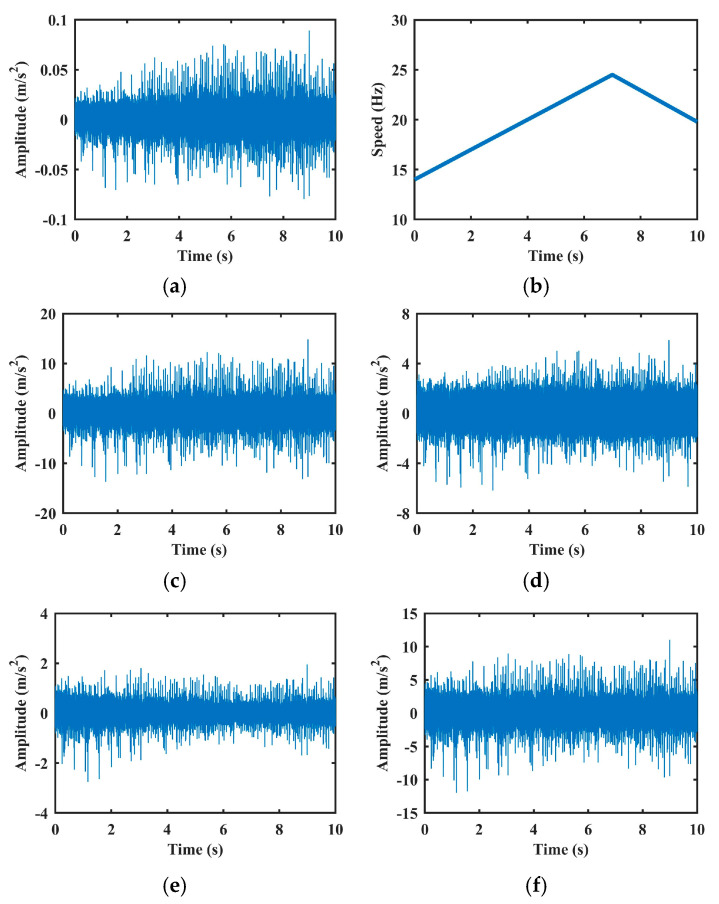
AM effect removal experiments of the vibration signal with inner fault in Case 2: (**a**) Raw vibration signal. (**b**) Rotation speed signal. (**c**–**f**) are results of methods [4,17,18], and the proposed method, respectively.

**Figure 13 sensors-25-06374-f013:**
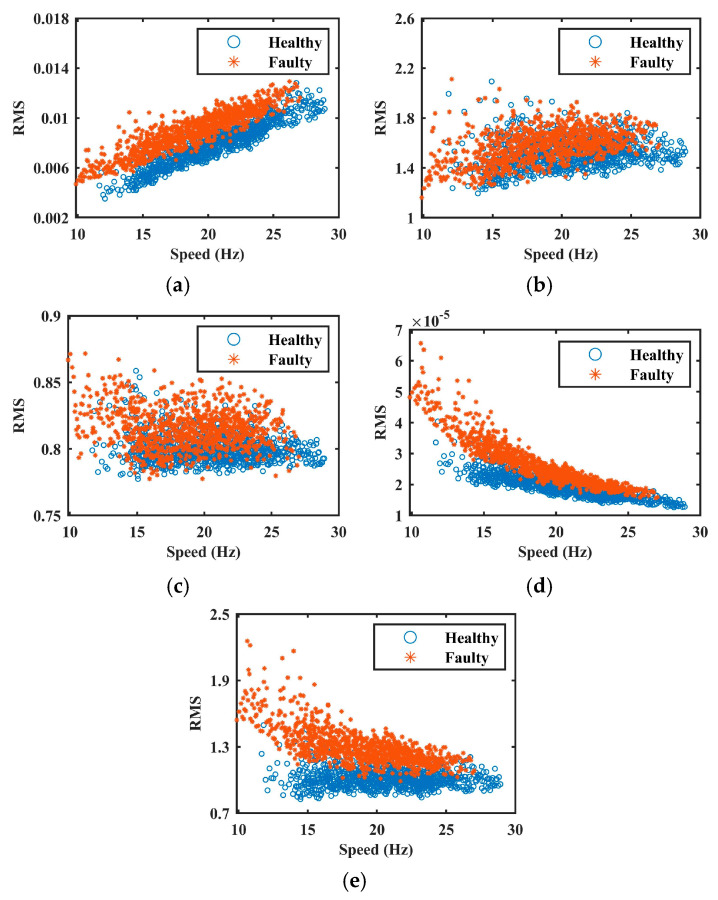
The health indices in Case 2. (**a**) raw signal. (**b**–**e**) are normalized signals by methods of [4,17,18], and the proposed method, respectively.

**Table 1 sensors-25-06374-t001:** The corresponding FSPC value for each feature vector in Case 1.

	Median	RMS	Peak-to-Peak	Kurtosis
FSPC	0.715231	0.618247	0.704482	−0.001362

**Table 2 sensors-25-06374-t002:** Fault detection results in Case 1.

Amplitude Normalization Method	Fault Detection AUC	
	Average	Standard Deviation
Raw signal	0.732	0.019
Method in Ref. [4]	0.776	0.024
Method in Ref. [17]	0.895	0.015
Method in Ref. [18]	0.853	0.018
Proposed method	0.985	0.009

**Table 3 sensors-25-06374-t003:** The corresponding FSPC value for each feature vector in Case 2.

	Median	RMS	Peak-to-Peak	Kurtosis
FSPC	0.923812	0.913843	0.571366	−0.214910

**Table 4 sensors-25-06374-t004:** Fault detection results in Case 2.

Amplitude Normalization Method	Fault Detection AUC	
	Average	Standard Deviation
Raw signal	0.913	0.021
Method in Ref. [4]	0.854	0.031
Method in Ref. [17]	0.945	0.015
Method in Ref. [18]	0.924	0.018
Proposed method	0.996	0.008

## Data Availability

The original contributions presented in this study are included in the article. Further inquiries can be directed to the corresponding author.

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
