# Peer review of "Amplitude Normalization for Speed-Induced Modulation in Rotating Machinery Measurements"

_sensors, 2025, doi:10.3390/s25206374_

Round 1

Reviewer 1 Report

Comments and Suggestions for Authors

this paper presents a technically sound and well-structured study on amplitude normalization for rotating machinery under variable speed. The use of SVR combined with multi-feature fusion is innovative and demonstrates strong improvements over traditional methods. With clearer emphasis on originality, broader benchmarking, and stronger discussion of real-world applicability, the paper could make a solid contribution to the field.

  1. The manuscript proposes an SVR-based amplitude normalization method for speed-induced modulation, which is novel compared to traditional envelope or quadratic fitting approaches. However, the authors should more clearly highlight how their contribution differs from and advances beyond recent deep-learning-based normalization methods.
  2. While the paper introduces a multi-feature fusion strategy, the selection of features (median, RMS, peak-to-peak, kurtosis) appears conventional. The authors could discuss the potential of incorporating more advanced time–frequency or nonlinear features to further strengthen the novelty.
  3. The overall organization of the manuscript is logical, progressing from introduction, methodology, and validation to conclusion. However, Section 2 (Theoretical background) could be more concise; some detailed derivations may be moved to an appendix.
  4. The contributions are well listed in the introduction, but they partly overlap with existing literature. The paper would benefit from a sharper articulation of “what is fundamentally new” in both the methodology and validation.

5.The comparative experiments with existing methods ([5], [18], [19]) are valuable, but the study lacks a comparison with more recent machine-learning or hybrid signal-processing approaches. This omission weakens the strength of the benchmarking.

  1. The SVR hyperparameter tuning relies on grid search, which is relatively simple. More justification is needed on why advanced optimization strategies (e.g., Bayesian optimization, evolutionary algorithms) are not employed, given their reported effectiveness.
  2. To strengthen the theoretical framework, the authors are suggested to incorporate recent relevant studies:

[1] Generalized synchroextracting transform: Algorithm and applications, Mechanical Systems and Signal Processing 224 (2025) 112116.

[2] CFFsBD: A Candidate Fault Frequencies-Based Blind Deconvolution for Rolling Element Bearings Fault Feature Enhancement. IEEE Transactions on Instrumentation and Measurement, 2023, 72: 3506412.

Author Response

Comment 1: The manuscript proposes an SVR-based amplitude normalization method for speed-induced modulation, which is novel compared to traditional envelope or quadratic fitting approaches. However, the authors should more clearly highlight how their contribution differs from and advances beyond recent deep-learning-based normalization methods.

Response 1: Thank you for this valuable suggestion. We agree that deep-learning-based normalization methods have been proposed. However, they typically require large datasets, involve high computational cost, and lack interpretability, which limit their practical use in machinery monitoring. By contrast, our SVR-based approach works well under small-sample conditions, is computationally efficient, and offers better interpretability. We have revised the literature review to include representative deep-learning studies, and highlighted how our contribution differs (line 74-80). We sincerely thank the reviewer for pointing out this omission, which has improved the completeness of our manuscript.

Comment 2: While the paper introduces a multi-feature fusion strategy, the selection of features (median, RMS, peak-to-peak, kurtosis) appears conventional. The authors could discuss the potential of incorporating more advanced time–frequency or nonlinear features to further strengthen the novelty.

Response 2: Thank you for this constructive suggestion. Since our work originates from an industrial project, the monitored signals mainly provide basic time-domain features such as median, RMS, peak-to-peak, and kurtosis. These features are simple, directly obtainable, and computationally efficient, which allows us to first examine whether satisfactory performance can be achieved with minimal complexity. We agree that incorporating more advanced time–frequency or nonlinear features could potentially enhance performance, but such approaches may also increase computational burden. We plan to address this in our future work by exploring richer feature sets and developing efficient algorithms that balance accuracy with real-time feasibility.

Comment 3: The overall organization of the manuscript is logical, progressing from introduction, methodology, and validation to conclusion. However, Section 2 (Theoretical background) could be more concise; some detailed derivations may be moved to an appendix.

Response 3: Thank you for the helpful comment. We agree that Section 2 was somewhat lengthy. To improve readability, we have revised this section by summarizing key points and condensing the expressions (the whole Section 2), so that the theoretical background is presented more concisely while keeping the overall structure intact.

Comment 4: The contributions are well listed in the introduction, but they partly overlap with existing literature. The paper would benefit from a sharper articulation of “what is fundamentally new” in both the methodology and validation.

Response 4: Thank you for this insightful comment. We have revised the contribution statements in the Introduction to sharpen the articulation of what is fundamentally new. Specifically, we emphasize that:

  1. Our work is the first to introduce a machine-learning approach (SVR) for modeling the nonlinear relationship between speed and amplitude modulation, in contrast to traditional envelope or functional approximations.
  2. We propose a multi-feature fusion strategy for more accurate quantification of AM effects, overcoming the limitations of single-feature methods.
  3. By combining these innovations, our method achieves markedly improved fault detection under variable-speed conditions.

These revisions clarify the novelty of both the methodology and its validation, as suggested by the reviewer (line 97-113).

Comment 5: The comparative experiments with existing methods ([5], [18], [19]) are valuable, but the study lacks a comparison with more recent machine-learning or hybrid signal-processing approaches. This omission weakens the strength of the benchmarking.

Response 5: Thank you for this professional suggestion. We fully agree that comparisons with more recent hybrid signal-processing methods can strengthen the benchmarking. In fact, this was precisely our motivation for including the study [18] as one of the comparative approaches. As described in Section 4.1.3, that method combines envelope analysis to extract the signal envelope with low-pass filtering to retain the speed-related components while preserving fault features, which represents a typical hybrid signal-processing strategy. We believe that incorporating this recent work has improved the scientific rigor of our experimental evaluation, and we sincerely appreciate the reviewer’s encouragement to further highlight this point.

Comment 6: The SVR hyperparameter tuning relies on grid search, which is relatively simple. More justification is needed on why advanced optimization strategies (e.g., Bayesian optimization, evolutionary algorithms) are not employed, given their reported effectiveness.

Response 6: Thank you for this helpful comment. We acknowledge that more advanced optimization strategies such as Bayesian optimization and evolutionary algorithms have been reported to improve SVR hyperparameter tuning. However, our main objective is to demonstrate the feasibility of eliminating speed-induced AM effects. Since even a basic grid search can already yield parameters that achieve accurate nonlinear fitting between speed and AM effects, more sophisticated optimization methods are expected to provide similar or better results without altering the core conclusion. This rationale has been noted in Section 3.2 when describing the SVR hyperparameter selection strategy.

Comment 7: To strengthen the theoretical framework, the authors are suggested to incorporate recent relevant studies:

[1] Generalized synchroextracting transform: Algorithm and applications, Mechanical Systems and Signal Processing 224 (2025) 112116.

[2] CFFsBD: A Candidate Fault Frequencies-Based Blind Deconvolution for Rolling Element Bearings Fault Feature Enhancement. IEEE Transactions on Instrumentation and Measurement, 2023, 72: 3506412.

Response 7: Thank you for this valuable suggestion. We have incorporated the recommended recent studies into the revised manuscript (Ref. [3] and [7]). These references have been added to strengthen the theoretical framework and improve the completeness of the literature review.

Reviewer 2 Report

Comments and Suggestions for Authors

This paper presents an amplitude normalization method based on Support Vector Regression to estimate and remove the AM effects. Overall this is an interesting topic and this manuscript is generally well written. However, the contribution appears to be mainly an application of machine learning techniques, with very limited advancement in underlying engineering knowledge. The authors are encouraged to provide reasonable insights into the physical mechanisms behind the phenomena, rather than relying solely on a combination of machine learning approaches, and treating the signals as a black box. 

The case studies are mainly on the monitoring and diagnosis of rolling element bearings under variable operational speed conditions. It should be noted that some well-established fault detection techniques, such as order tracking and envelope analysis, have been widely published and accepted in this research area. Authors should provide a detailed benchmarking against these conventional and widely used methods, and clearly demonstrate the advantages and added values of their proposed approach.

It is unclear whether the authors conducted any experimental work themselves, or if the analysis is based solely on datasets by other researchers. Clarification on this point would strengthen the paper’s credibility.

A careful proof reading is recommended to correct some minor language issues. 

In Eq. (3), please clarify how the speed sequence is constructed or obtained. 

Page 6: what is the length of the moving average window? How was it determined?

Some figures are too blur. 

Author Response

We sincerely thank the reviewer for the positive recognition of our work. The physical mechanism of AM is that the variation of signal amplitude with changes in operating conditions such as speed. To capture these variations, we employ multiple time-domain features that reflect how the amplitude evolves over time. However, applying envelope extraction or recalculating AM effects for each signal individually would significantly increase computational cost, which is impractical for industrial monitoring.

Based on this understanding, our approach focuses on identifying the underlying relationship between speed and amplitude variation. By learning this nonlinear mapping once—using signals and corresponding speed data collected under healthy conditions—we can subsequently apply the trained model directly to new signals, thus enabling real-time AM suppression while retaining fault-related information. This design is not a purely “black-box” application of machine learning, but rather a practical implementation rooted in the physical mechanism of speed-induced modulation and the demands of industrial feasibility.

To further clarify this motivation, we have revised and streamlined the theoretical background section to better explain the mechanism of speed-induced AM effects (the whole Section 2), thereby highlighting the physical basis of our method in addition to the machine learning implementation.

Comment 1: The case studies are mainly on the monitoring and diagnosis of rolling element bearings under variable operational speed conditions. It should be noted that some well-established fault detection techniques, such as order tracking and envelope analysis, have been widely published and accepted in this research area. Authors should provide a detailed benchmarking against these conventional and widely used methods, and clearly demonstrate the advantages and added values of their proposed approach.

Response 1: Thank you for this insightful comment. We fully agree that order tracking and envelope analysis are well-established techniques in fault detection under variable-speed conditions. However, it should be clarified that order tracking is primarily designed to address frequency modulation problems, and is therefore not applicable to the amplitude modulation effects that are the focus of this study. For this reason, order tracking was not included in our case study comparisons.

Similarly, envelope analysis is often used in conjunction with order tracking to construct envelope order spectra for frequency-modulation-based diagnosis. In the context of amplitude modulation, however, envelope analysis may not accurately estimate the AM effect. As discussed in the Introduction, its limitations in this regard motivated the development of alternative normalization strategies. To ensure a fair comparison, we included in our benchmarking the method of Ref. [5], which is an envelope-analysis-based approach, thereby representing this class of conventional techniques.

Comment 2: It is unclear whether the authors conducted any experimental work themselves, or if the analysis is based solely on datasets by other researchers. Clarification on this point would strengthen the paper’s credibility.

Response 2: Thank you for raising this question. We would like to clarify that our study is not solely based on external datasets. In particular, the dataset used in Case Study 1 was collected by the authors through experiments conducted in the laboratory of Taiyuan University of Technology. We have revised the manuscript to explicitly state this point in the dataset description to avoid misunderstanding (line 333). We believe this clarification strengthens the credibility of our work.

Comment 3: A careful proof reading is recommended to correct some minor language issues.

Response 3: Thank you for this suggestion. We have carefully proofread the manuscript and corrected the minor language issues.

Comment 4: In Eq. (3), please clarify how the speed sequence is constructed or obtained.

Response 4: Thank you for the valuable comment. To clarify, the speed sequence in Eq. (3) is constructed by discretizing the continuous speed signal , which is measured by a shaft-mounted encoder. Specifically, at each sampling instant corresponding to the vibration signal acquisition, the instantaneous rotational speed is recorded. This process yields the discrete sequence , which is then used in the subsequent modeling. We have added this explanation to the manuscript to make the construction of the speed sequence clearer (line 201).

Comment 5: Page 6: what is the length of the moving average window? How was it determined?

Response 5: Thank you for the question. The moving average window should not be too short, because it must capture fluctuations in the vibration signal. At the same time, it should not be too long, so that the speed within the window can still be regarded as approximately stable. In this study, we set the window length to 1/500 of the total signal duration, which provides a sufficient number of data points while maintaining local stationarity. We have added this clarification to the manuscript (line 234-240).

Comment 6: Some figures are too blur.

Response 6: Thank you for pointing this out. We have checked the figures and confirmed that they are in JPEG format. The blurriness may have been caused by compression during the PDF conversion process. We will carefully adjust the export settings and ensure that all figures are clear in the revised submission.

Reviewer 3 Report

Comments and Suggestions for Authors

My comments to enhance the structure of the manuscript are:

1) Please do not define an abbreviation if it is not used at least twice in the manuscript (e.g., COT). Please also not define the same abbreviation multiple times (e.g., RMS).

2) The authors have two subsections under Section 2. Still, both have the same numbering (2.1). In other words, you have “two” Section 2.1. In addition, Section 2 (as a whole) can be enhanced. The current version is limited. For example, Section 2.2 (you have incorrectly numbered it) is more like an extension regarding the criticism of the state-of-the-art. It does not sound like “Theoretical Background.”

3) This reviewer liked the criticism and discussion of the available literature. Still, I have a question about using the following terms while defining your approach: “innovative” and “new.”

Many studies use Pearson’s correlation for different purposes with slight changes. Your SVR-based approach seems to be a combination of other existing techniques. Please justify these arguments if you are willing to present them as a new metric (Line 202) or innovative (Line 12). In other words, are you proposing something really new, or are you implementing slightly adjusted (but existing) methods on a new application area?

4) Please also add the pertinent reference numbers to the labels of Figures 4 and 9. What does AN stand for in Figure 6?

5) Focus on enhancing the graphic quality of the figures. It is hard to read the text.

6) Are you using all the data for a specific scenario to train your model? For example, are you using all the data points for the healthy state, or only a specific data segment?

What is your input for different tasks? It is unclear (e.g., Lines 351 and 453). Can you clearly define the term “faulty?” Do you mean the bearing data with the inner raceway fault for all cases? If not, are you gathering all the faulty conditions (outer raceway, multi-fault, etc.) in one class? If so, what is the rationale behind it?

Why did the authors specifically select the “inner raceway” fault?

Please consider evaluating other performance metrics besides the “AUC.”

How did you calculate the average value and the standard deviation values? What is the number of trials? For example, did you run your model five times and then calculate the average values? Also, please use “standard deviation” instead of “Std.” as you have enough space in the table.

The same questions apply to Case 2.

Line 466: “…can be accessed via DOI: 10.17632/v43hmbwxpm.1.” Providing the DOI number in the text is unnecessary. Please remove it. Citing the relevant reference is enough.

Please use the minus sign instead of the hyphen while indicating negative values (e.g., Line 486).

Author Response

Comment 1: Please do not define an abbreviation if it is not used at least twice in the manuscript (e.g., COT). Please also not define the same abbreviation multiple times (e.g., RMS).

Response 1: Thank you for pointing this out. We have carefully checked the manuscript to ensure that abbreviations are only defined when used more than once, and that each abbreviation (e.g., RMS) is defined only once.

Comment 2: The authors have two subsections under Section 2. Still, both have the same numbering (2.1). In other words, you have “two” Section 2.1. In addition, Section 2 (as a whole) can be enhanced. The current version is limited. For example, Section 2.2 (you have incorrectly numbered it) is more like an extension regarding the criticism of the state-of-the-art. It does not sound like “Theoretical Background”.

Response 2: Thank you for the careful observation. We have corrected the numbering issue to ensure that each subsection under Section 2 is properly labeled. In addition, we have revised Section 2 to strengthen its role as the theoretical background. Specifically, we condensed redundant derivations in Section 2.1 and reorganized the content of the former Section 2.2. The latter now provides a clearer theoretical foundation on speed-induced amplitude modulation, rather than appearing only as a critique of existing methods. These revisions improve both the accuracy of presentation and the logical flow of the paper (the whole Section 2).

Comment 3: This reviewer liked the criticism and discussion of the available literature. Still, I have a question about using the following terms while defining your approach: “innovative” and “new”.

Many studies use Pearson’s correlation for different purposes with slight changes. Your SVR-based approach seems to be a combination of other existing techniques. Please justify these arguments if you are willing to present them as a new metric (Line 202) or innovative (Line 12). In other words, are you proposing something really new, or are you implementing slightly adjusted (but existing) methods on a new application area?

Response 3: Thank you for this important comment. We agree that Pearson’s correlation coefficient is a well-established tool and not a new metric in itself. In our work, it is employed as a criterion to construct a multi-feature set correlated with speed, which is then used in the SVR-based normalization framework. The novelty lies in the integration of correlation-guided feature selection with multi-feature fusion and SVR modeling for speed-induced AM suppression, rather than in the correlation measure itself.

To avoid misunderstanding, we have revised the manuscript in several places: line 12, line 104, and line 193. These changes provide a more accurate description of our approach and ensure that the manuscript does not unintentionally overstate the originality of Pearson’s correlation itself.

Comment 4: Please also add the pertinent reference numbers to the labels of Figures 4 and 9. What does AN stand for in Figure 6?

Response 4: Thank you for the helpful comment. We have revised the captions of Figures 4 and 9 to clarify that these experimental setup diagrams are taken from the corresponding references, and we have added the appropriate reference numbers in the captions (line 344 and 477). In Figure 6, “AN” originally denoted the learned nonlinear relationship. We apologize for the confusion caused by the unclear labeling. To improve readability, we have replaced “AN” with “Normalization Function” in both the figure and the text (line 383 and 502).

Comment 5: Focus on enhancing the graphic quality of the figures. It is hard to read the text.

Response 5: Thank you for this comment. We have enhanced the graphic quality of all figures, increased the resolution, and adjusted the font size to ensure that the text and details are clearly readable in the revised manuscript.

Comment 6: Are you using all the data for a specific scenario to train your model? For example, are you using all the data points for the healthy state, or only a specific data segment?

Response 6: Thank you for this important question. In our study, we used all of the healthy-state data for training. The reason is that the training samples need to cover the full range of operating speed variations. Only by doing so can the model learn a reliable normalization function that is valid across the entire speed interval. We noticed that this point was not clearly explained in the original manuscript, which may have caused confusion for readers. We have added this clarification in the revised version (line 226).

Comment 7: What is your input for different tasks? It is unclear (e.g., Lines 351 and 453). Can you clearly define the term “faulty?” Do you mean the bearing data with the inner raceway fault for all cases? If not, are you gathering all the faulty conditions (outer raceway, multi-fault, etc.) in one class? If so, what is the rationale behind it? Why did the authors specifically select the “inner raceway” fault?

Response 7: Thank you for raising this important question. Our experiments consist of two parts, and the inputs differ accordingly. In the first part, we focus on learning the normalization function. Here, the SVR is trained exclusively with vibration signals collected under healthy operating conditions, ensuring that the learned function captures only the speed–AM relationship without being influenced by fault-related information. After the normalization function is obtained, it is applied to all signals in the dataset, including both healthy and faulty cases (line 346).

In the second part, we evaluate the impact of normalization on fault detection performance. For this task, an AE model is used, with the input training set consisting of normalized healthy and faulty signals. This enables the AE to learn representative features of both conditions and perform fault detection (line 445).

Regarding the definition of “faulty,” in the first part we used inner race fault data only as an example to illustrate the effect of different normalization methods, but this does not mean that only inner race faults were considered. The dataset also contains outer race and mixed fault conditions. In the second part, we treat all faulty conditions (inner race, outer race, and mixed faults) as a single “fault” class. This is because our task is fault detection rather than fault classification. From a practical perspective, once a fault is detected—regardless of type—the equipment must be inspected. Fault detection is also the fundamental prerequisite for more fine-grained classification, which we plan to explore in future work.

We acknowledge that the original manuscript did not explain these points clearly, which may have caused confusion. We have revised the relevant sections to provide a clearer description (line 346 and 445).

Comment 8: Please consider evaluating other performance metrics besides the “AUC.”

Response 8: Thank you for this constructive suggestion. We selected AUC as the evaluation metric because it simultaneously considers both true positive and false positive rates and provides a robust and widely accepted measure of detection performance, especially under varying operating conditions. Since our focus in this work is on demonstrating the effectiveness of amplitude normalization rather than comparing classifiers, we considered AUC sufficient to reflect the relative improvements across different methods. We agree that incorporating additional performance metrics (e.g., accuracy, precision, recall, F1-score) would provide a more comprehensive evaluation, and we will consider these in our future work.

Comment 9: How did you calculate the average value and the standard deviation values? What is the number of trials? For example, did you run your model five times and then calculate the average values? Also, please use “standard deviation” instead of “Std.” as you have enough space in the table.

The same questions apply to Case 2.

Response 9: Thank you for the valuable comment. In the fault detection experiments, we first processed the entire dataset using each amplitude normalization method. The processed signals were then input into the AE model for fault detection. To ensure reliability, each experiment was repeated three times, and the reported results represent the average values over these trials. We have added this clarification in the revised manuscript (line 464). In addition, we have replaced “Std.” with the full term “standard deviation” in the tables (line 468 and 520).

Comment 10: Line 466: “…can be accessed via DOI: 10.17632/v43hmbwxpm.1.” Providing the DOI number in the text is unnecessary. Please remove it. Citing the relevant reference is enough.

Response 10: Thank you for pointing this out. We have removed the DOI number from the main text and kept only the citation to the corresponding reference (line 470).

Comment 11: Please use the minus sign instead of the hyphen while indicating negative values (e.g., Line 486).

Response 11: Thank you for this careful observation. We have corrected the formatting and now use the proper minus sign instead of the hyphen when indicating negative values throughout the manuscript (line 495).

Round 2

Reviewer 3 Report

Comments and Suggestions for Authors

Thank you for incorporating the reviewer's comments.

1) "...features with FSPC values below a threshold (0.5 in this study) are discarded to prevent degradation of model performance." If possible, add a reference here to support this argument.

- Lines 455 to 458: "Model training was conducted using Adam optimizer with...." Are these optimized values? Or, did the authors adopt these hyperparameters from other studies? Or, did you obtain them through brute force? The authors optimized the hyperparameters for their amplitude normalization model using grid search. However, their selections regarding the fault detection model are unclear.

2) Lines 360 to 368: The authors still use hyphens instead of the minus sign while indicating the negative values.

Author Response

Comment 1: "...features with FSPC values below a threshold (0.5 in this study) are discarded to prevent degradation of model performance." If possible, add a reference here to support this argument.

Response 1: Thank you for this helpful comment. In this study, the threshold of 0.5 for the FSPC was chosen as an empirical criterion. In the statistical literature, a Pearson correlation coefficient of |r| ≥ 0.5 is generally interpreted as a moderate-to-strong correlation, indicating a meaningful association between two variables [Schober, P., Boer, C., & Schwarte, L. A.. Correlation coefficients: appropriate use and interpretation. Anesthesia & Analgesia, 2018, 126(5), 1763–1768.].

Furthermore, we only retain features with positive correlations above 0.5. This is because, from the physical mechanism of speed-induced AM effects, the vibration amplitude is expected to increase with rotational speed due to higher excitation energy, impact intensity, and friction forces. Features that exhibit negative correlation with speed do not conform to this mechanism and are therefore discarded as unreliable descriptors of speed-induced modulation.

We have added this clarification and the corresponding references in the revised manuscript (line 248 to 253).

Comment 2: Lines 455 to 458: "Model training was conducted using Adam optimizer with...." Are these optimized values? Or, did the authors adopt these hyperparameters from other studies? Or, did you obtain them through brute force? The authors optimized the hyperparameters for their amplitude normalization model using grid search. However, their selections regarding the fault detection model are unclear.

Response 2: Thank you for the insightful comment. The AE was constructed as a baseline framework to evaluate the contribution of amplitude normalization to fault detection. Therefore, we did not perform extensive hyperparameter optimization for the AE model. Instead, we adopted widely used default settings reported in the literature, namely the Adam optimizer with an initial learning rate of 0.001 [Kingma D P. Adam: A method for stochastic optimization. arxiv preprint arxiv:1412.6980, 2014. doi: 10.48550/arXiv.1412.6980], a batch size of 64, and a maximum of 100 epochs, which are consistent with configurations commonly employed in autoencoder-based signal processing and fault diagnosis studies. These settings are known to provide stable convergence and reliable performance, and they ensure that the comparison fairly reflects the impact of different normalization methods rather than differences in classifier tuning.

We have added this clarification in the revised manuscript (line 458 to 463).

Comment 3: Lines 360 to 368: The authors still use hyphens instead of the minus sign while indicating the negative values.

Response 3: Thank you for the suggestion. We have corrected the formatting and replaced all hyphens with proper minus signs when indicating negative values in the revised manuscript (line 363, 370, 494 and 498).
